# Anastomotic Leak in Ovarian Cancer Cytoreduction Surgery: A Systematic Review and Meta-Analysis

**DOI:** 10.3390/cancers14215464

**Published:** 2022-11-07

**Authors:** Massimiliano Fornasiero, Georgios Geropoulos, Konstantinos S. Kechagias, Kyriakos Psarras, Konstantinos Katsikas Triantafyllidis, Panagiotis Giannos, Georgios Koimtzis, Nikoletta A. Petrou, James Lucocq, Christos Kontovounisios, Dimitrios Giannis

**Affiliations:** 1Medical School, University College London, London WC1E 6BT, UK; 22nd Propaedeutic Department of Surgery, Aristotle University School of Medicine, Hippokration General Hospital, 546 42 Thessaloniki, Greece; 3Department of General and Upper GI Surgery, Victoria Hospital Kirkcaldy, Kirkcaldy KY2 5AH, UK; 4Society of Meta-Research and Biomedical Innovation, London W12 0FD, UK; 5Department of Metabolism, Digestion and Reproduction, Faculty of Medicine, Imperial College London, London SW7 2AZ, UK; 6Department of Life Sciences, Faculty of Natural Sciences, Imperial College London, London SW7 2AZ, UK; 7Public Medical Centre, Krya Vrysi, 58300 Giannitsa, Greece; 8Department of General Surgery, The Royal Marsden Hospital, London SW3 6JJ, UK; 9Department of Hepaticopancreaticobiliary Surgery, University of Edinburgh, Edinburgh EH8 9YL, UK; 10Department of Surgery, North Shore University Hospital/Long Island Jewish Medical Center, Northwell Health, Manhasset, NY 11030, USA; 11Donald and Barbara Zucker School of Medicine at Hofstra/Northwell, Hempstead, NY 11549, USA

**Keywords:** anastomotic leak, ovarian cancer, cytoreductive surgery, systematic review, meta-analysis

## Abstract

**Simple Summary:**

Bowel resection is often required to obtain complete removal of ovarian cancer. A major complication of this operation is anastomotic leakage, which has been shown to increase morbidity and mortality in this population. Numerous original research studies have assessed the risk factors for anastomotic leaks. We aimed to conduct a systematic review and meta-analysis to identify statistically significant risk factors. This meta-analysis identified multiple bowel resections as the only significant risk factor. With further research to identify additional risk factors, new management guidelines could be implemented to minimize the risk of anastomotic leaks and improve patient outcomes.

**Abstract:**

Introduction: Anastomotic leaks (AL) following ovarian cytoreduction surgery could be detrimental, leading to significant delays in commencing adjuvant chemotherapy, prolonged hospital stays and increased morbidity. The aim of this study was to investigate risk factors associated with anastomotic leaks after ovarian cytoreduction surgery. Material and methods: The MEDLINE (via PubMed), Cochrane Library, EMBASE and Scopus bibliographical databases were searched. Original clinical studies investigating risk factors for AL in ovarian cytoreduction surgery were included. Results: Eighteen studies with non-overlapping populations reporting on patients undergoing cytoreduction surgery for ovarian cancer (n = 4622, including 344 cases complicated by AL) were included in our analysis. Patients undergoing ovarian cytoreduction surgery complicated by AL had a significantly higher rate of 30-day mortality but no difference in 60-day mortality. Multiple bowel resections were associated with an increased risk of postoperative AL, while no association was observed with body mass index (BMI), American Society of Anesthesiologists (ASA) score, age, smoking, operative approach (primary versus interval cytoreductive, stapled versus hand-sewn anastomoses and formation of diverting stoma), neoadjuvant chemotherapy and use of hyperthermic intraperitoneal chemotherapy (HIPEC). Discussion: Multiple bowel resections were the only clinical risk factor associated with increased risk for AL after bowel surgery in the ovarian cancer population. The increased 30-day mortality rate in patients undergoing ovarian cytoreduction complicated by AL highlights the need to minimize the number of bowel resections in this population. Further studies are required to clarify any association between neoadjuvant chemotherapy and decreased AL rates.

## 1. Introduction

Ovarian cancer remains the most lethal gynecological malignancy, with a five-year survival of 43% in the United Kingdom [1]. Optimal cytoreductive surgery, resulting in no residual disease, is the mainstay of treatment. Patients with advanced ovarian cancer, who have optimal cytoreduction, have prolonged 5-year survival compared to patients who have residual disease greater than 1 cm [2]. Therefore, ultra-radical surgery is often undertaken. These procedures traditionally involve resection of other abdominopelvic tissue that has likely been invaded by primary ovarian cancer [3]. In addition to surgery, the use of hyperthermic intraperitoneal chemotherapy (HIPEC) and neoadjuvant or adjuvant chemotherapy have also been shown to improve survival rates [4].

Bowel resection is frequently required to achieve optimal cytoreduction, particularly in advanced ovarian cancer [4]. The most common type of bowel resection performed in this context is rectosigmoid resection [5]. Primary bowel anastomosis, compared to permanent stoma formation, is the preferred method of repair following bowel resection [6]. A commonly reported severe complication of bowel anastomosis is an anastomotic leak (AL), which occurs in 8–14% of patients [7,8] and is associated with significant morbidity and mortality [9].

Given the detrimental effects of AL on patient outcomes, the identification of risk factors for AL is of utmost importance to improve prognosis in the ovarian cancer population. The risk factors associated with AL in patients undergoing colorectal surgery have been well described in the literature and include a prolonged operating time, the use of neoadjuvant chemotherapy, anastomotic level (i.e., low versus high rectal anastomosis) and pre-operative hypoalbuminemia [10,11,12]. However, given the surgical and pathological differences between ovarian and bowel cancer, the risk factors for AL are expected to differ in ovarian cancer cytoreduction surgery. This study aims to synthesize the most current primary evidence to identify the preoperative and intra-operative risk factors for AL in the ovarian cancer population undergoing cytoreductive surgery with bowel resection and anastomosis.

## 2. Materials and Methods

### 2.1. Study Design and Inclusion/Exclusion Criteria

This systematic review and meta-analysis were conducted according to the Preferred Reporting Items for Systematic Reviews and Meta-analyses (PRISMA) guidelines and in line with the protocol developed and agreed a priori by all authors. Studies investigating risk factors for anastomotic leaks in patients undergoing ovarian cytoreduction surgery were deemed eligible for analysis. Exclusion criteria were: (i) articles published in languages other than English, (ii) narrative or systematic reviews and meta-analyses, (iii) case reports, errata, comments, perspectives, letters to the editor and editorials that did not provide any extractable data, (iv) published abstracts with no available full text and (v) non-comparative studies (single-arm studies). No publication date, sample size restrictions or any other search filters were applied. This study is registered with PROSPERO, number CRD42022364076.

### 2.2. Search Strategy

Eligible studies were identified by searching through the MEDLINE (via PubMed), Cochrane Library, Embase and Scopus databases (end-of-search date: 28 November 2021) by two independent researchers. The search strategies used are described in more detail in Appendix A. Any disagreements were resolved by a third reviewer. The reference lists and all previously published systematic reviews were thoroughly searched for missed studies eligible for inclusion based on the “snowball” methodology [13].

### 2.3. Data Extraction

A standardized, pre-piloted form was used for data tabulation and extraction. Two reviewers extracted the data independently, and any disagreements were identified and resolved by a third reviewer. We extracted the following data from the included studies: (i) study characteristics (first author, year of publication, study design, study center, country, study period and number of patients), (ii) patient characteristics (BMI, smoking, age and American Society of Anesthesiologists (ASA) physical status classification), (iii) operation (primary, secondary), (iv) intraoperative outcomes (type of anastomosis, stoma formation), (v) mortality outcomes and (vi) chemotherapy administration (bevacizumab, HIPEC).

### 2.4. Risk of Bias Assessment

We assessed the risk of bias using the Risk of Bias tool of the National Heart, Lung and Blood Institute (NHLBI). The tool examines eight domains as possible sources of bias: (i) study objectives, (ii) study population, (iii) consecutiveness of the population, (iv) comparability of the subjects, (v) intervention, (vi) measurement of outcomes, (vii) follow-up and (viii) statistical analysis results. For each domain, questions are answered with “yes” or “no”. Based on these answers, an overall risk of bias assessment was calculated for each included study [14].

### 2.5. Statistical Analysis

Available data were handled according to the principles stated in the Cochrane Handbook [13]. Data on outcomes of interest were summarized and analyzed cumulatively. Categorical variables were reported as the number of events among the total cases. Based on the extracted data, the odds ratio (OR) and 95% confidence interval (CI) were calculated by means of 2 × 2 tables for categorical events; OR > 1 indicated that the trait was more frequently present in the AL group. Between-study heterogeneity was assessed by estimating the I2 statistic. Continuous variables were summarized as means and standard deviations (SDs). Weighted mean differences (WMD) and 95% CIs were estimated for each continuous outcome; WMD > 0 corresponded to larger values in the normal group. High heterogeneity was confirmed with a significance level of *p* < 0.05 and I2 ≥ 50%. The random-effects model was used to calculate the pooled effect when heterogeneity was high, while the fixed-effects model was used when low heterogeneity was encountered. All statistical analyses and forest plots were performed with the use of Reviewer Manager 5.4.1 software (Review Manager (RevMan) [Computer Program]. Version 5.4.1, Copenhagen: The Nordic Cochrane Centre, Denmark, The Cochrane Collaboration, 2020).

## 3. Results

### 3.1. Study Selection, Study Characteristics

Through our systematic search, 125 unique articles were retrieved, 68 of which underwent full-text evaluation for eligibility (57 studies excluded based on title and abstract screening). Ultimately, 18 studies reporting on 4622 patients undergoing cytoreduction surgery for ovarian cancer (4278 non-AL and 344 AL patients) fulfilled the inclusion criteria and were included in our quantitative data synthesis [8,15,16,17,18,19,20,21,22,23,24,25,26,27,28,29,30,31] (Figure 1) [32]. Six of the studies were conducted in the United States of America, four in Germany, three in South Korea, one in the United Kingdom, one in Austria, one in Spain, one in France, and one was multicentric. All studies were retrospective. Baseline patient characteristics, including ovarian tumor histology, staging and CC-0 resection rate, are described in Table 1. Inclusion and exclusion criteria as well as the definition of anastomotic leak among the included studies, are summarized in Table 2.

### 3.2. Patient Characteristics

The FIGO staging system was used in the included studies. Eight studies included only those patients with ovarian cancer of stage 3 or above [6,15,16,17,19,22,29,30]. Five studies included patients of stage 2 and above [18,21,23,24,31]. Four studies included patients with all stages of ovarian cancer [8,25,26,27]. One study did not provide staging information [20]. Mean BMI (WMD 0.61, 95% CI: −0.74 to 1.96, *p* = 0.38, I2 = 0%) and current smoking status (OR: 0.60, 95% CI: 0.29 to 1.24, *p* = 0.12, I2 = 0%) were not associated with a significantly increased risk of AL, as reported by two and three studies, respectively [21,25]. Mean age at the time of surgery was not significantly associated with AL (WMD 0.76, 95% CI: −1.47 to 3.00, *p* = 0.5, I2 = 40%) (Figure 2). Neither the ASA I-II classification (OR: 1.42, 95% CI: 0.69 to 2.91, *p* = 0.34, I2 = 0%) nor the ASA III-IV classification (OR: 1.39, 95% CI: 0.94 to 2.04, *p* = 0.10, I2 = 0%) were significantly associated with AL (Figure 3).

### 3.3. Neoadjuvant, Bevacizumab and HIPEC Therapy

The use of neoadjuvant chemotherapy was associated with a statistically non-significant decrease in the rate of AL (OR 0.58, 95% CI: 0.33 to 1.02, *p* = 0.06, I2 = 44%). Similarly, in subgroup analysis, there was no difference in AL rate with the use of bevacizumab (OR: 0.78, 95% CI: 0.29 to 2.09, *p* = 0.63, I2 = 57%). The use of HIPEC was also not linked to a statistically significant increase in AL rates (OR: 1.42, 95% CI: 0.69 to 2.91, *p* = 0.34, I2 = 0%) when combined with surgical management (Figure 4).

### 3.4. Intraoperative Considerations

The number of anastomoses carried out intraoperatively (OR: 1.60, 95% CI: 1.06 to 2.40, *p* = 0.02, I2 = 0%) was associated with an increase in AL. The type of anastomosis, including hand-sewn (OR: 1.39, 95% CI: 0.79 to 2.44, *p* = 0.25, I2 = 36%) or stapled (OR: 1.09, 95% CI: 0.70 to 1.71, *p* = 0.7, I2 = 53%) was not associated with a significant increase in AL (Figure 5). Protective stoma (OR: 0.82, 95% CI: 0.51 to 1.30, *p* = 0.39, I2 = 69%) as well as primary (OR: 1.72, 95% CI: 0.94 to 3.15, *p* = 0.08, I2 = 22%) versus secondary cytoreductive or recurrent disease were not associated with a significantly altered risk of AL (Figure 6).

### 3.5. Mortality

AL was significantly associated with an increase in 30-day mortality (OR: 2.51, 95% CI: 1.13 to 5.57, *p* = 0.02, I2 = 0%) but not with 60-day mortality (OR: 1.88, 95% CI: 0.80 to 4.43, *p* = 0.15, I2 = 40%) (Figure 7).

### 3.6. Risk of Bias Assessment

The mean score of the NHLBI scale was 7.7 ± 1.1. These results highlight that the included studies were, on average, of good quality. This is outlined in Table 3.

## 4. Discussion

This systematic review and meta-analysis of 18 retrospective studies showed that multiple bowel resections increased the risk of AL following ovarian cytoreduction surgery. In addition, the presence of AL in ovarian cytoreduction surgery significantly increased the risk of 30-day mortality. These findings indicate that multiple bowel resections may be a major contributor to poor outcomes, including 30-day mortality, in patients with ovarian cancer who require bowel anastomosis. Historically, AL was thought to only impact 30-day mortality [33,34], given the well-established risk of associated postoperative morbidity [35] and reoperation following diagnosis [36]. However, this hypothesis is unlikely to accurately reflect the true extent of the physiological burden associated with AL based on recent studies recording higher absolute values of clinically, but not statistically, significant 60-day and 90-day mortality (11.1% to 19.1%) [23,36]. In support of these findings, 60-day mortality was not significantly increased following AL in our meta-analysis. Given that a key consequence of AL is a delay in starting adjuvant therapy [37], a study with long-term follow-up may be required to fully assess the long-term consequences of AL on morbidity and mortality and the safe interval of initiation of adjuvant chemotherapy in the setting of AL.

Our analysis revealed that the use of neoadjuvant chemotherapy was associated with a clinically significant decrease in the rate of AL that marginally failed to attain statistical significance. In the neoadjuvant setting, a typical regimen consists of two or three courses administered prior to interval cytoreductive surgery, followed by up to six courses of chemotherapy postoperatively [38]. This regimen is advocated for in the guidelines published by the National Comprehensive Cancer Network (NCCN) for patients who are poor surgical candidates or have a low likelihood of successful cytoreduction [39]. The European Society for Medical Oncology (ESMO) guidelines remain equivocal [40]. The aim of neoadjuvant chemotherapy is to downstage tumors with a low chance of optimal cytoreductivity with upfront surgery [16]. Randomized controlled trials have shown non-inferiority in terms of survival for patients receiving neoadjuvant therapy compared to primary surgical management [41,42]. Further analysis completed by Kehoe et al. [41] demonstrated that neoadjuvant therapy significantly reduced the risk of Clavien Dindo grade III and IV complications compared to adjuvant therapy as per the National Cancer Institute Common Terminology Criteria for Adverse Events. In contrast, a meta-analysis demonstrated decreased survival in individuals receiving neoadjuvant chemotherapy compared to those receiving upfront surgery [43]. This, in part, can be explained through the requirement of neoadjuvant chemotherapy only in those with advanced cancer where suboptimal surgical cytoreduction and reduced overall survival are anticipated.

Our analysis further confirmed that there is no significant difference in AL rates in patients receiving HIPEC. HIPEC consists of circulating warmed chemotherapeutic agents around the peritoneal cavity intraoperatively, following resection and prior to anastomosis or stoma formation, and aims to eliminate any remaining neoplastic cells. A recent phase III trial, randomizing patients with advanced ovarian cancer to either HIPEC or no HIPEC during cytoreductive surgery, demonstrated that HIPEC increases recurrence-free survival and overall survival [44]. Moreover, there was no increase in side effects. These findings demonstrated that the addition of HIPEC is safe and efficacious in patients with advanced ovarian cancer. Of note, the aforementioned study included only 24 individuals that underwent bowel resection, with the majority receiving a protective colostomy. This raises concerns regarding the potential for adverse events associated with bowel resection following HIPEC treatment in patients with advanced ovarian cancer. Subsequent analysis by Gruner et al. [7] investigated the incidence and associated risk factors for anastomotic failure following interval cytoreductive surgery with or without HIPEC and showed that there was no significant difference in AL rates between those who underwent interval cytoreductive surgery with or without HIPEC. These data are consistent with our findings and support the hypothesis that the anticipated risk of AL should not guide the use of HIPEC in ovarian cancer patients.

Protective stoma formation describes the process of creating a temporary diverting proximal stoma in patients at higher risk of complications, such as AL, to prevent postoperative morbidity [45]. Our analysis did not show any statistically significant difference in AL rates of patients undergoing the creation of a protective stoma. Koscielny et al. [8] found a statistically significant reduction in the AL rate, but this effect was not confirmed by larger studies. This overall result is in part due to the heterogeneity between studies; criteria for covering stoma formation vary between studies and between individual surgeons, and overall, the patient numbers are small. For example, most stomas reported in studies were created at the surgeon’s discretion based on the fulfilment of certain criteria. While Koscielny et al. [8] generated a stoma if expected blood loss exceeded 1000 mL, Grimm et al. [15] did not use a blood loss cut-off as a criterion. Protective stoma reduces the rate of AL, as well as the length of stay in the hospital and postoperative mortality [36]. A meta-analysis of 27 studies showed an association between a defunctioning stoma with significantly decreased AL rates in patients with rectal cancer [46]. However, a recent systematic review and meta-analysis of 17 studies with a total number of 2719 patients by Santana et al. revealed that protective stoma formation did not decrease AL rates in ovarian cancer cases, thus suggesting that its use is limited to only selected cases [47]. Finally, a study of 145 patients that underwent colorectal resection during cytoreduction surgery for ovarian cancer reported no differences in AL rates among patients with or without diverting ileostomy and ghost ileostomy [48].

Secondary debulking describes cytoreduction surgery in the setting of recurrence in patients that underwent primary surgery to remove the tumor found in the first place. Our analysis found no significant difference in AL rates between individuals undergoing primary or secondary recurrent disease cytoreduction procedures. However, in the literature, the eligibility criteria for secondary cytoreductive surgery seem to be more selective than for primary cytoreductive surgery [49,50]. Therefore, secondary cytoreductive surgery patients tend to be fitter for surgery and are at a reduced risk of postoperative complications prior to the cytoreduction surgery. Optimal cytoreductive surgery in residual disease is a matter of controversy. Harter et al. [50] suggested an improved prognosis only if complete resection can be achieved, while another meta-analysis demonstrated benefits for patients with microscopic residual disease [51].

The limitations of this study include the use of observational studies as opposed to randomized controlled trials. Other limitations include the small number of patients, with few studies primarily designed to investigate AL-associated risk factors, and many of these only consider AL in subgroup analysis. Furthermore, additional relevant factors, including the surgeon’s experience, blood loss and the need for transfusion, were not amenable to analysis due to limited available data. The role of preoperative hypoalbuminemia (albumin level < 3.4 g/dL) was not examined in this study as it is a well-established independent preoperative risk factor for AL after colorectal surgery [52,53,54,55] and more recently, a 2022 systematic review and meta-analysis of 3274 patients demonstrated that a preoperative albumin level of <3.0 g/dL is also a significant risk factor of AL after bowel resection and anastomosis for ovarian cancer [56,57]. The research regarding the AL rate for bowel cancer patients is more extensive, potentially offering valuable insight for further reduction of AL in the ovarian cancer population [52].

## 5. Conclusions

In conclusion, our study suggests that AL is associated with a significantly increased 30-day mortality rate in the ovarian cancer population undergoing a cytoreduction operation. Multiple bowel resections were a risk factor for AL in this patient population. However, larger prospectively-designed studies are required to increase the statistical power of any future analyses and more accurately assess a wider range of risk factors for an anastomotic leak in ovarian cytoreduction, similar to bowel cancer studies.

## Figures and Tables

**Figure 1 cancers-14-05464-f001:**
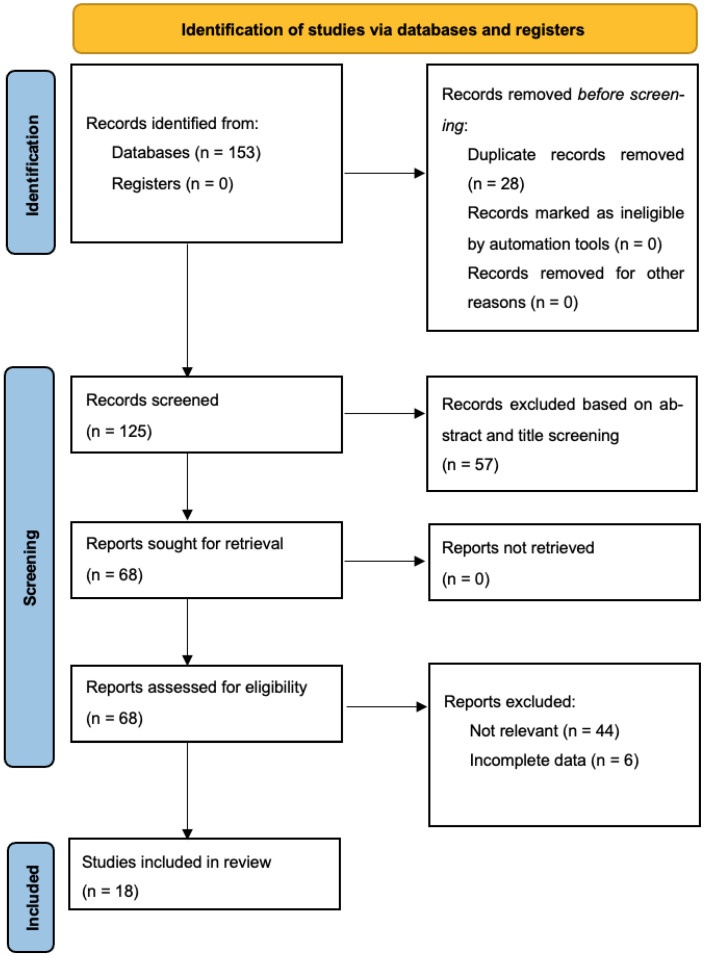
PRISMA flowchart.

**Figure 2 cancers-14-05464-f002:**
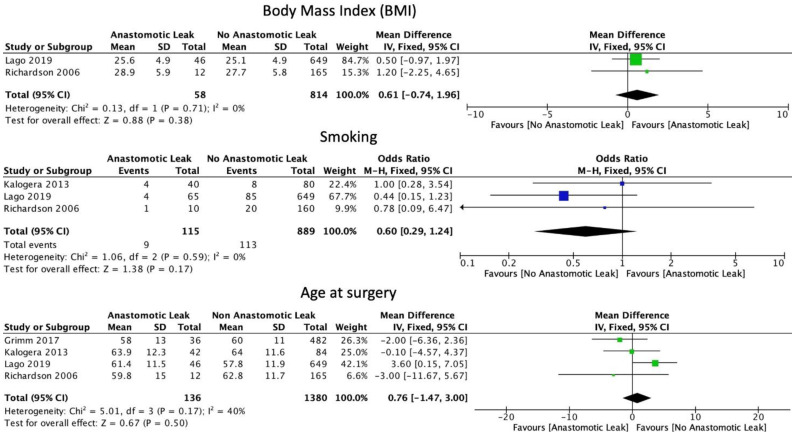
Basic patient characteristics.

**Figure 3 cancers-14-05464-f003:**
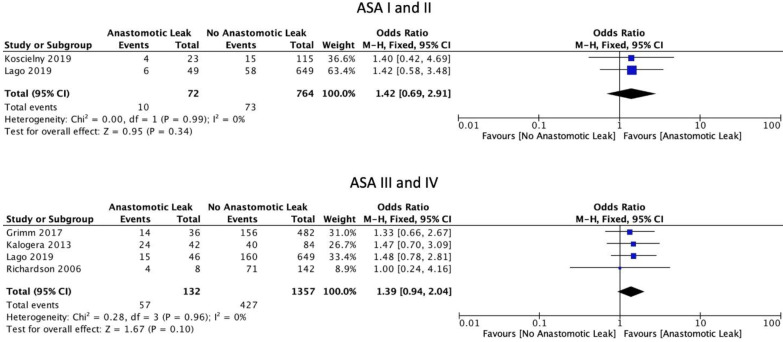
ASA scores.

**Figure 4 cancers-14-05464-f004:**
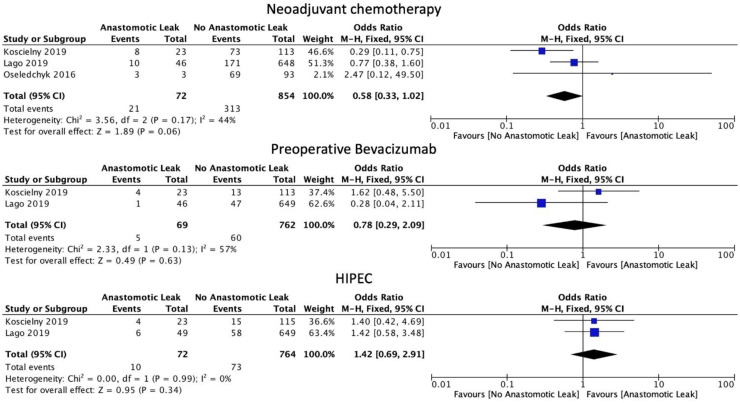
Chemotherapy administration.

**Figure 5 cancers-14-05464-f005:**
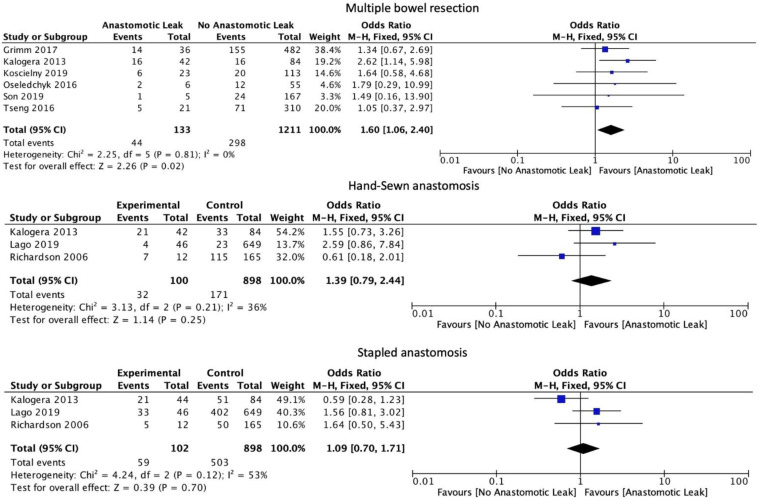
Intraoperative outcomes.

**Figure 6 cancers-14-05464-f006:**
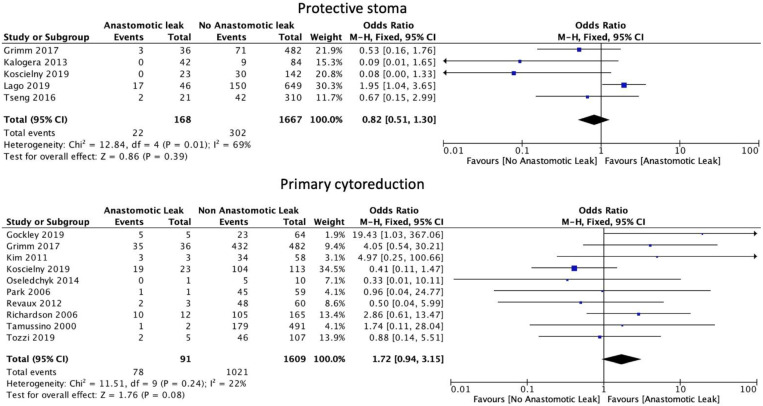
Stoma formation and primary cytoreduction risk factors.

**Figure 7 cancers-14-05464-f007:**
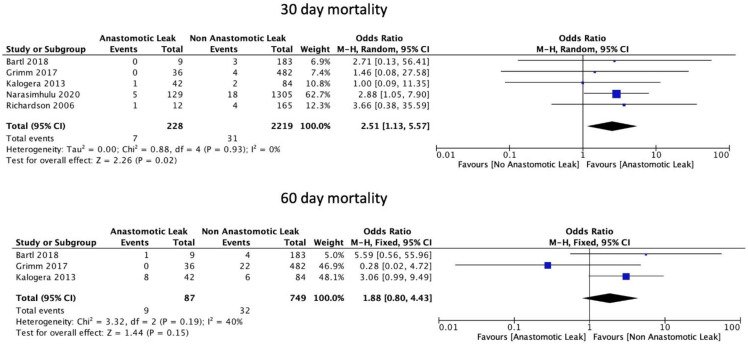
Mortality rates.

**Table 1 cancers-14-05464-t001:** Basic characteristics of the included studies. Anastomotic leak (AL), Epithelial (E), Malignant Mixed Mullerian Tumor (MMT), Granulosa cell (G).

Study (Year)	Country	Period	Sample Size (n)	Number of AL Patients (%)	Histology, n (type)	FIGO Staging, n (% of Sample Size) (Stage)	Number of CC-0 Resections (%)
Richardson et al. (2006)	USA	January 1999 to December 2004	177	12 (7%)	167 (E) 7 (MMT) 3 (G)	3 (1.7) (I) 11 (6.2) (II) 128 (72.3) (III) 26 (14.6) (IV) 9 (5.1) (Unknown)	73 (41%)
Lago et al. (2019)	Multicentric	January 2010 to June 2018	695	46 (7%)	572 (Serous) 49 (Endometrioid) 15 (Mucinous) 22 (Clear Cell) 36 (Other)	29 (4.2) (II) 418 (60.1) (III) 114 (16.4) (IV) 134 (19.3) (Unknown)	
Tseng et al. (2016)	USA	January 2005 to January 2014	331	21 (6%)	292 (Serous) 5 (Endometrioid) 4 (Clear cell) 30 (Other)	11 (3.3) (II) 231 (69.8) (III) 89 (26.9) (IV)	
Bartl et al. (2018)	Austria	Between 2003 and 2017	192	9 (5%)	193 (Epithelial ovarian cancer)	154 (80.2) (IIB-IIIC) 38 (19.8) (IV)	90 (47%)
Son et al. (2019)	South Korea	January 2006 to January 2018	172	5 (3%)	146 (Serous) 21 (Non-serous) 5 (Non-epithelian ovarian cancer)	128 (74.4) (IIIc or IV)	
Tozzi et al. (2019)	United Kingdom	January 2009 to March 2016	112	5 (5%)	83 (Serous) 29 (Others)	81 (72.3) (IIIc) 29 (27.7) (IV)	
Oseledchyk et al. (2016)	Germany	Between 2002 to 2013	96	3 (3%)	91 (Serous) 2 (Mucinous) 1 (MMT) 1 (Clear cell) 1 (Intestinal)	75 (78.1) (IIIc) 21 (21.9) (IV)	0 (0%)
Oseledchyk et al. (2014)	Germany	January 2005 to September 2013	11	1 (9%)	10 (Epithelial) 1 (MMT)	9 (81.8) (III) 2 (18.1) (IV)	7 (64%)
Estes et al. (2006)	USA	1996 to 2001	48	1 (2%)	24 (Papillary) 11 (Endometrioid) 8 (Mixed) 2 (Mucinous) 2 (Unkown) 1 (Clear cell)	1 (2.1) (IIIa) 3 (6.3) (IIIb) 42 (87.5) ((IIIc) 2 (4.2) (IV)	14 (29%)
Grimm et al. (2017)	Germany	January 1999 to December 2015	518	36 (7%)	518 (Serous)	262 (50.6) (III) 256 (49.4) (IV)	469 (59%)
Kalogera et al. (2013)	USA	January 1994 to May 2011	126 (AL and 1:2 matched controls)	42 (33%)	NG	12 (9.5) (IIIa/b) 81 (64.3) (IIIc) 33 (26.2) (IV)	11 (91%)
Kim et al. (2011)	South Korea	January 1998 to August 2008	61	3 (5%)	61 (Epithelial, of which 25 are serous)	51 (83.6) (III) 10 (16.4) (IV)	35 (57%)
Koscielny et al. (2019)	Germany	2010 to 2017	136	23 (17%)	136 Epithelial	104 (76.5) (I/II) 30 (22.1) (III/IV)	
Lago et al. (2018)	Spain	December 2016 to December 2017	26	2 (8%)	22 (Serous) 2 (MMMT) 2 (Undifferentiated)	7 (26.9)) (IIb) 1 (3.8) (IIIb) 12 (46.1) (IIIc) 4 (15.4) (IVa) 2 (7.7) (Not applicable—relapse)	
Park et al. (2006)	South Korea	April 2001 to May 2005	60	1 (2%)	47 (Serous) 13 (Other)	2 (3.3) (I) 1 (1.7) (II) 54 (90) (III) 3 (5) (IV)	50 (83%)
Revaux et al. (2012)	France	2001 to 2009	63	3 (5%)	39 (Serous) 12 (Endometrioid) 2 (Mucinous) 4 (Papillary) 6 (Other)	5 (7.9) (II) 52 (82.5) (III) 6 (9.5) (IV)	51 (81%)
Tamussino et al. (2000)	USA	January 1983 to December 1992	364	2 (1%)	306 (Serous) 17 (Mucinous) 39 (Endometrioid) 8 (Clear cell) 72 (Undifferentiated) 43 (Other)	29 (8.0) (I) 31 (8.5) (II) 254 (69.8) ((III) 50 (13.7) (IV)	216 (59%)
Narasimhulu et al. (2020)	USA	January 2012 to December 2016	1434	129 (9%)			

**Table 2 cancers-14-05464-t002:** Definition of anastomotic leak among the included studies as well as inclusion/exclusion criteria.

Study (Year)	Inclusion Criteria	Exclusion Criteria	Anastomotic Leak Definition
Richardson et al. (2006)	Debulking surgery which included a rectosigmoid resection primary or recurrent ovarian or primary peritoneal cancer	Patient having end colostomy, diverting stoma, and inadequate postoperative follow-up after hospital discharge to assess bowel integrity	Evidence of fecal drainage from deep drains, wound or vagina, and/or evidence of either extravasation of contrast at the anastomotic site or evidence of communication between the rectum and pelvic abscess noted on radiographic imaging.
Lago et al. (2019)	Cytoreductive surgery for primary advanced or relapsed ovarian cancer with colorectal resection and anastomosis	Patients with end colostomy or end ileostomy, as well as those for whom relevant information was missing	
Tseng et al. (2016)	Stage II to IV ovarian, fallopian tube, or primary peritoneal carcinoma who underwent large bowel resection during PDS	Patients excluded if they had received neoadjuvant chemotherapy prior to attempted primary debulking surgery	
Bartl et al. (2018)	Patients with International Federation of Gynecology and Obstetrics (FIGO) advanced Epithelial ovarian carcinoma stage (IIB-IV) who underwent primary cytoreductive surgery	Patients with missing data about primary therapy, with missing informed consent, with recurrent disease, or with other concomitant malignancies were excluded from the study	
Son et al. (2018)	Patients with advanced ovarian cancer who underwent bowel resection as part of debulking surgery	patients who did not undergo full circumference bowel resection or those who underwent only appendectomy during the debulking procedure	Anastomotic leakage was defined as the drainage of fecal fluid or extravasations of anastomosis sites verified with computed tomography and the patients’ clinical symptoms
Tozzi et al. (2019)	Stage IIIC–IV ovarian cancer patients who had bowel surgery with rectosigmoid resection		
Oseledchyk et al. (2016)	Patients with macroscopic tumor residuals after surgery for advanced-stage ovarian cancer were included	Patients who were operated on for borderline tumors, nonepithelial histology, or for recurrent disease were excluded	
Oseledchyk et al. (2014)	All patients who had undergone surgery including a total or subtotal colectomy as part of cytoreductive surgery for primary or recurrent ovarian cancer.		
Estes et al. (2006)	Stage III or IV epithelial ovarian cancer who had bowel resection as part of primary cytoreductive surgery.	Patients diagnosed with ovarian cancer undergoing bowel resection to avoid obstruction	
Grimm et al. (2017)	Patients with primary advanced high-grade serous epithelial ovarian cancer (stage III-IV) undergoing primary or interval debulking surgery.	All patients who did not undergo full circumference bowel resection or received an appendectomy only	Feculent fluid from drains; vaginal vault or wound; extravasation from anastomotic site verified by computed tomography and/or leakage confirmed at revision surgery.
Kalogera et al. (2013)	All patients who underwent large bowel resection with primary anastomosis during cytoreductive surgery for primary or recurrent ovarian cancer.	Excluded from the analysis given the lack of appropriately matched control patients.	Feculent fluid from drains, wound or vagina; definitive radiographic evidence of extravasation at the anastomotic site or leakage found at reoperation.
Kim et al. (2011)	Advanced primary epithelial ovarian cancer with histological confirmation; stage III-IV disease; Eastern Co-operative Oncology Group performance status 0–2; primary cytoreductive surgery including low anterior resection and anastomosis or Hartmann’s procedure; adjuvant treatment with taxane and platinum-based chemotherapy after primary surgery and no neoadjuvant chemotherapy For patients with recurrent epithelial ovarian cancer inclusion criteria were: platinum sensitivity; up to three resectable metastatic tumors at preoperative evaluation; low anterior resection and anastomosis or Hartmann’s procedure for cytoreduction of recurrent tumors; no previous bowel surgery at rectosigmoid colon; up to three previous chemotherapeutic regimens and performance status 0–2		
Koscielny et al. (2019)	All patients who underwent any type of bowel resection and primary anastomosis during debulking surgery of confirmed ovarian cancer.	Patients without full circumferential bowel resection; Hartmann’s procedure or other discontinuity resection without anastomosis; later diagnosis of histology other than ovarian cancer; diffuse and deep infiltration of the small bowel mesentery root; diffuse carcinomatosis of the small bowel involving such large resections that resection would result in short bowel syndrome; diffuse involvement of stomach/duodenum without the possibility of limited resection, or of the head or middle part of the pancreas; tumor involvement of truncus coeliacus, hepatic arteries or left gastric artery; with central or multiple liver and pulmonary metastases.	Feculent secretion from drains, wound or vagina, extravasation from an anastomotic site verified by computed tomography, air exiting from drains during diagnostic rectoscopy or leakage confirmed at revision surgery.
Lago et al. (2018)	Patients who underwent bowel resection as part of modified pelvic exenteration in the treatment of ovarian cancer.		
Park et al. (2006)	Patients with advanced (stage IIIb-IV) ovarian cancer undergoing low anterior en bloc resection as part of cytoreductive surgery, followed by adjuvant chemotherapy with taxane and platinum		
Revaux et al. (2012)	Patients undergoing modified posterior pelvic exenteration for advanced ovarian cancer, either as part of primary cytoreductive surgery or interval cytoreductive surgery.		
Tamussino et al. (2000)	Patients who underwent one or more operations for ovarian cancer, including a gastrointestinal procedure by gynecological surgeons.	Patients who underwent appendectomy.	
Narasimhulu et al. (2020)	Patients who underwent colon resection as part of complex cytoreductive surgery	Patients with ASA > 4; ventilator dependence; open wound; acute renal failure; undergoing dialysis; sepsis within 48 h prior to surgery and those undergoing emergent surgery.	

**Table 3 cancers-14-05464-t003:** The risk of bias assessment using the Risk of Bias tool of the National Heart, Lung and Blood Institute (NHLBI). Y = Yes. N = No.

**First Author**	Publication Year	Was the Study Question or Objective Clearly Stated?	Was the Study Population Clearly and Fully Described, Including a Case Definition?	Were The Cases Consecutive?	Were the Subjects Comparable?	Was the Intervention Clearly Described?	Were the Outcome Measures Clearly Defined, Valid, Reliable, and Implemented Consistently across All Study Participants?	Was the Length of Follow-Up Adequate?	Were the Statistical Methods Well-Described?	Were the Results Well-Described?	Total Score	Quality Rating
C. Grimm	2017	N	Y	Y	Y	Y	Y	Y	Y	Y	8	High Quality
A. Koscielny	2019	Y	Y	Y	Y	Y	Y	Y	Y	Y	9	High Quality
V. Lago	2019	Y	N	Y	Y	N	Y	Y	Y	Y	7	High Quality
A. Oseledchyk	2014	Y	Y	Y	Y	Y	Y	Y	N	Y	8	High Quality
A. Oseledchyk	2016	Y	N	Y	Y	Y	Y	Y	Y	Y	8	High Quality
A. Revaux	2012	Y	N	Y	Y	Y	Y	Y	Y	Y	8	High Quality
D. Richardson	2016	Y	Y	Y	Y	Y	Y	Y	Y	Y	9	High Quality
J.H. Son	2019	Y	Y	Y	N	Y	Y	Y	N	Y	7	High Quality
R. Tozzi	2019	Y	N	Y	Y	Y	Y	Y	Y	Y	8	High Quality
J. Tseng	2016	Y	Y	Y	N	Y	Y	Y	Y	Y	8	High Quality
T. Bartl	2018	Y	Y	Y	Y	Y	Y	Y	Y	Y	9	High Quality
J. Estes	2006	Y	Y	Y	N	Y	Y	Y	Y	Y	8	High Quality
E. Kalogera	2013	Y	Y	Y	N	Y	Y	Y	Y	Y	8	High Quality
H. Kim	2011	Y	Y	Y	Y	Y	N	Y	Y	Y	8	High Quality
V. Lago	2018	Y	Y	Y	Y	Y	N	Y	Y	Y	8	High Quality
J.Y. Park	2006	Y	Y	Y	N	Y	Y	Y	Y	Y	8	High Quality
K. Tamussino	2000	Y	Y	Y	N	Y	Y	Y	Y	Y	8	High Quality
D. Narasimhulu	2020	Y	Y	Y	N	Y	N	Y	Y	Y	7	High Quality

## Data Availability

Any data reported have been appropriately referenced, with no original data being reported by the authors.

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
