# Peer review of "Anastomotic Leak in Ovarian Cancer Cytoreduction Surgery: A Systematic Review and Meta-Analysis"

_cancers, 2022, doi:10.3390/cancers14215464_

Round 1
Reviewer 1 Report
-Interpretation and presentation of previous studies is accurate.
-No major suggestions for improvements.
-Clarity and context in this paper are good.
-References are comprehensive.
Author Response
Response to Reviewer 1 Comments
-Interpretation and presentation of previous studies is accurate.
-No major suggestions for improvements.
-Clarity and context in this paper are good.
-References are comprehensive.
Response 1: Dear Reviewer, we would like to thank you for your feedback.
Reviewer 2 Report
This paper is a review and meta-analyze about anastomotic leak (AL) after ovarian cancer surgery with 4622 patients. The aim of the paper is to determine pre- and intra-operative risk factors for AL. Its main contributions are the patient number and the pre- and intra-operative risk factor study together.
This article is about a clinical question significant. The methodology is correct, after corrections and precisions. This papier is relevant with, until today, a unique review published about this question. The quality must be check after search strategy precision. The findings are of interest to physicians caring for ovarian cancer. The proposed study has more patients than the recently published review.
Thirteen non-randomised studies in Valenti et al. against eighteen in this paper, including 8 studies considered in both.
Valenti, G., Vitagliano, A., Morotti, M., Giorda, G., Sopracordevole, F., Sapia, F., Lo Presti, V., Chiofalo, B., Forte, S., Lo Presti, L., & Tozzi, R. (2022). Risks factors for anastomotic leakage in advanced ovarian cancer: A systematic review and meta-analysis. Eur J Obstet Gynecol Reprod Biol, 269, 3-15. https://doi.org/10.1016/j.ejogrb.2021.12.007
A risk factor for AL is, in this published review and in some studies, pre albumin level ≤3 gr/dL, it’s necessary to discuss this point in the discussion.
The main modification to be made in this article are clarifications on the methodology: the name of the databases queried must be harmonized between Abstract and M and M. (scopus or Embase?). It would be very relevant to have the Search Strategy in the supplementary data. Regarding this subject, the MEDLINE database is certainly the most relevant, interesting articles could be appended in Embase and not in MEDLINE, which is why the precision of the methodology is important.
NB: the MEDLINE database includes the COCHRANE; so, querying both databases is unnecessary.
Some specific corrections would benefit:
+L49 standard treatment is “complete cytoreduction”, so “complete cytoreductive surgery” must replace “debulking surgery”, standard treatment is CC-0. The surgery « optimal » is define until 2,5 tumoral residual!
Brennan, D. J., & Moran, B. J. (2021). Time to Evolve Terminology from "Debulking" to Cytoreductive Surgery (CRS) in Ovarian Cancer. Ann Surg Oncol, 28(11), 5805-5807. https://doi.org/10.1245/s10434-021-10490-4
du Bois, A., Reuss, A., Pujade-Lauraine, E., Harter, P., Ray-Coquard, I., & Pfisterer, J. (2009). Role of surgical outcome as prognostic factor in advanced epithelial ovarian cancer: a combined exploratory analysis of 3 prospectively randomized phase 3 multicenter trials: by the Arbeitsgemeinschaft Gynaekologische Onkologie Studiengruppe Ovarialkarzinom (AGO-OVAR) and the Groupe d'Investigateurs Nationaux Pour les Etudes des Cancers de l'Ovaire (GINECO). Cancer, 115(6), 1234-1244. https://doi.org/10.1002/cncr.24149
+L89 scopus database is not EMBASE
+L307 title without text??!
+L473 2 times the number 20
+Table, the description of the tumor burden via the Peritoneal Carcinosis Index to describe the populations would be relevant.
+Table 1 Number of R0 Resections (%) could be replaced by Number of CC-0 resections
+Table 1 To Oseledchyk MMMT could be replaced by MMT
+Table 2, Not Given is no useful
+Table 2 D Richardson 2016 and not 20016
+Figure 1: Details for the 57 records excluded would be interesting
FIGO stage, number of patients with the % would be nicer to compare the series.
The English language is appropriate.
Author Response
Response to Reviewer 2 Comments
This paper is a review and meta-analyze about anastomotic leak (AL) after ovarian cancer surgery with 4622 patients. The aim of the paper is to determine pre- and intra-operative risk factors for AL. Its main contributions are the patient number and the pre- and intra-operative risk factor study together.
This article is about a clinical question significant. The methodology is correct, after corrections and precisions. This papier is relevant with, until today, a unique review published about this question. The quality must be check after search strategy precision. The findings are of interest to physicians caring for ovarian cancer. The proposed study has more patients than the recently published review.
Thirteen non-randomised studies in Valenti et al. against eighteen in this paper, including 8 studies considered in both.
Valenti, G., Vitagliano, A., Morotti, M., Giorda, G., Sopracordevole, F., Sapia, F., Lo Presti, V., Chiofalo, B., Forte, S., Lo Presti, L., & Tozzi, R. (2022). Risks factors for anastomotic leakage in advanced ovarian cancer: A systematic review and meta-analysis. Eur J Obstet Gynecol Reprod Biol, 269, 3-15. https://doi.org/10.1016/j.ejogrb.2021.12.007
Response 1: Dear Reviewer, we would like to thank you for your feedback. A recently published review attempted to answer a similar question to our review. They searched up to October 2020 whereas we searched up until November 2021. This may account for the greater number of studies included in our analysis. Given the increased number of included studies, we were able to quantitatively analyse more potential risk factors for anastomotic leak including patient characteristics (BMI, age, ASA grade) and treatment characteristics (use of chemotherapy and/or bevacizumab, use of hyperthermic intraperitoneal chemotherapy).
Point 1: The main modification to be made in this article are clarifications on the methodology: the name of the databases queried must be harmonized between Abstract and M and M. (scopus or Embase?). It would be very relevant to have the Search Strategy in the supplementary data. Regarding this subject, the MEDLINE database is certainly the most relevant, interesting articles could be appended in Embase and not in MEDLINE, which is why the precision of the methodology is important.
Response 1: The description of the databases searched has been made more consistent. All four databases were searched (Embase, Medline, Cochrane and Scopus). We have added to our manuscript the search strategy as supplementary information.
Point 2: +L49 standard treatment is “complete cytoreduction”, so “complete cytoreductive surgery” must replace “debulking surgery”, standard treatment is CC-0. The surgery « optimal » is define until 2,5 tumoral residual!
Response 2: Dear reviewer, we would like to thank you for the feedback. We totally agree that the term cytoreduction is more appropriate. We have proceeded to the replacement of debulking with the cytoreduction term throughout our manuscript.
Point 3: +L89 scopus database is not EMBASE
Response 3: Dear reviewer, we would like to thank you for the feedback. We used four databases for our systematic search. We have ammended the material and methods clarifying that the MEDLINE (via PubMed), Cochrane Library, and Scopus databases were used.
Point 4: +L307 title without text??!
Response 4: Dear reviewer, many thanks for the observation. We have removed the title as inserted by mistake.
Point 5: +L473 2 times the number 20
Response 5: The second 20 has been removed
Point 6: +Table, the description of the tumor burden via the Peritoneal Carcinosis Index to describe the populations would be relevant.
Response 6: Dear reviewer, we would like to thank you for these valuable observations. Indeed Peritoneal Carcinosis Index (PCI) could provide useful information about the reported population. However, the included studies do not report such sufficient data to assist a table formation with the PCI variable.
Point 7: +Table 1 Number of R0 Resections (%) could be replaced by Number of CC-0 resections
Response 7: The number of R0 resections has been replaced by the number of CC-0 resections
Point 8: +Table 1 To Oseledchyk MMMT could be replaced by MMT
Response 8: MMMT has been replaced by MMT
Point 9: +Table 2, Not Given is no useful
Response 9: “Not given” has been removed
Point 10: +Table 2 D Richardson 2016 and not 20016
Response 10: The additional “0” has been removed
Point 11: +Figure 1: Details for the 57 records excluded would be interesting
Response 11: Dear reviewer, many thanks for this comment. These 57 studies were excluded in the tittle and abstract screening process stage. The following sentence was added in the PRISMA flow chart and results part of our manuscript: (57 studies excluded based on title and abstract screening)
Point 12: FIGO stage, number of patients with the % would be nicer to compare the series.
Response 12: The % of patients with each stage as a proportion of the sample size for each study has been added in
Round 2
Reviewer 2 Report
ok with this modifications